# Sensor Location Matters When Estimating Player Workload for Baseball Pitching

**DOI:** 10.3390/s22229008

**Published:** 2022-11-21

**Authors:** Cristine Agresta, Michael T. Freehill, Jessica Zendler, Georgia Giblin, Stephen Cain

**Affiliations:** 1Department of Rehabilitation Medicine, University of Washington, Seattle, WA 98195, USA; 2Department of Orthopedic Surgery, Stanford University, Redwood City, CA 94063, USA; 3Rimkus Consulting Group, Woodinville, WA 9807, USA; 4Detroit Tigers, Inc., Detroit, MI 48201, USA; 5Department of Chemical and Biomedical Engineering, University of West Virginia, Morgantown, WV 26506, USA

**Keywords:** load, stress, torque, wearables, arm

## Abstract

Estimating external workload in baseball pitchers is important for training and rehabilitation. Since current methods of estimating workload through pitch counts and rest days have only been marginally successful, clubs are looking for more sophisticated methods to quantify the mechanical loads experienced by pitchers. Among these are the use of wearable systems. While wearables offer a promising solution, there remains a lack of standards or guidelines for how best to employ these devices. As a result, sensor location and workload calculation methods vary from system to system. This can influence workload estimates and blur their interpretation and utility when making decisions about training or returning to sport. The primary purpose of this study was to determine the extent to which sensor location influences workload estimate. A secondary purpose was to compare estimates using different workload calculations. Acceleration data from three sensor locations—trunk, throwing upper arm, and throwing forearm—were collected from ten collegiate pitchers as they threw a series of pitches during a single bullpen session. The effect of sensor location and pitch type was assessed in relation to four different workload estimates. Sensor location significantly influenced workload estimates. Workload estimates calculated from the forearm sensor were significantly different across pitch types. Whole-body workload measured from a trunk-mounted sensor may not adequately reflect the mechanical loads experienced at throwing arm segments. A sensor on the forearm was the most sensitive to differences in workloads across pitch types, regardless of the calculation method.

## 1. Introduction

Estimating external workload in baseball pitchers is important for training and rehabilitation. To build strength and resilience in musculoskeletal tissues, it is necessary to overload the tissue using a magnitude that promotes positive adaptation [1]. However, if the magnitude of loading exceeds tissue tolerance, it can cause injury. Repetitive cycles of loading without adequate recovery time can also negatively impact soft tissue structures and increase susceptibility to injury, particularly for high-velocity movements [1]. 

Traditional strategies to manage workload and reduce the negative effects of tissue overload use limits on pitch counts to reduce the total number of loading cycles or rest: work schedules to allow for tissue recovery between loading [2]. In recent years, these strategies have been applied to both youth and professionals. However, they have had little success in mitigating overuse injuries [3,4]. Since current methods of managing workload through pitch counts and rest days have had only marginal success [5], clubs are looking toward more sophisticated methods. Among these are the use of wearables to estimate and track workload. Anecdotally, player tracking using commercially available wearable devices is becoming more commonplace in baseball organizations. There has also been an uptick in the number of publications using wearable devices to predict injury [6], evaluate training programs to reduce elbow stress [7], identify pitching mechanics that increase susceptibility to injury [8,9,10], and determine which pitches generate the largest external workload [11]. All studies have used a single sensor either on the medial side of the proximal forearm [6,7,8,9,10] or at the base of the neck on the trunk [11]. While wearables offer a promising solution to the lack of precision in current load management practices, there are currently no standards or guidelines for how to best employ these devices to obtain robust measures. As a result, sensor location and calculation methods vary from system to system. This may influence workload estimates and make interpretation difficult, which in turn limits the utility of these estimates to assist in training or rehabilitation decision-making.

The primary purpose of this study was to determine if sensor location influenced workload estimates. We hypothesized that workload estimates would differ across sensor locations. Our secondary purpose was to compare four different workload calculations typically used with acceleration data—peak resultant acceleration, peak PlayerLoad, cumulative PlayerLoad, and normalized resultant acceleration.

## 2. Materials and Methods

This study was approved by the Institutional Review Board. All participants provided informed consent before data collection.

### 2.1. Participants

Ten male D1 collegiate pitchers (left-handed = 4, mean age = 19.2 ± 1.2 years, height = 186.4 ± 6.2 cm, mass = 86.5 ± 10.5 kg) participated in this study. Pitchers were cleared for practice and were free of injury. 

### 2.2. Data Collection and Analysis

We used an observational approach and collected data simultaneously from sensors on three body segments—throwing forearm, throwing upper arm, and trunk—while pitchers threw a series of pitches during their typical bullpen session. Data was collected during the preseason with regular season games (or scrimmages) scheduled for several months away. 

Each pitcher wore five inertial sensors (Opal, APDM, Inc., Portland, OR, USA). Each sensor incorporated a tri-axial accelerometer (range: ±200 g) and a tri-axial gyroscope (range: ±2000°/s) sampled at 512 Hz. Sensors were placed on the dorsum of each foot, the throwing forearm, the throwing upper arm, and the trunk (Figure 1). Sensors were affixed prior to the start of warm-up to allow for a seamless transition of the pitcher’s individual warm-up throws into pitching. Sensor data collection was initiated immediately prior to the start of the warm-up and collected continuously until the end of the bullpen session. A time marker was added to the data stream during collection to denote the transition from warm-up to bullpen session. The warm-up times ranged between 9 and 22 min, with an average warm-up time of approximately 12 ± 3.6 min. Pitchers completed a similar, but not identical warm-up process. All pitchers followed their own preferred warm-up routine consisting of jogging, stretches, and throwing. 

Pitchers threw from regulation distance on the same regulation mound in their team indoor cage. Pitchers threw a series of approximately 35 pitches to a live catcher in a pre-determined order set by the pitching coach. Each pitcher threw about 18 fastballs, 7 change-ups, and 10 breaking balls (curveballs, sliders, or cutters) (Table 1). Pitchers were instructed to throw at their full effort, since data were collected in preseason (October) and pitchers had been throwing for several weeks. Five pitchers threw 1–4 additional pitches beyond the set order per self-request or on the coach’s recommendation. These pitches were included in data analysis. Ball kinematics were captured using a consumer-available radar unit (Pitching 2.0, Rapsodo, Inc., Brentwood, MO, USA). In addition, one pitch type per player was recorded via high-speed video (S-MOTION, AOS Technologies, Plymouth, MI, USA, 500 Hz). Two cameras were used to capture video: one camera placed in front of the mound facing the pitcher and the other camera placed on the pitcher’s throwing arm side. High-speed video was utilized in post-processing to verify segment motion, if needed. 

Pitches were identified in the data by using both sensor and radar unit data. The magnitude of the acceleration of each segment was calculated. A custom Matlab (MATLAB, Mathworks, Natick, MA, USA) program was used to identify throws from the upper arm resultant acceleration. The program searched for peak values of acceleration in magnitudes greater than 200 m/s^2^ that were separated by a minimum of 2 s. Identified pitches in the sensor data were matched with corresponding pitches recorded by the radar system by using the time stamps of the identified throws (sensor data) and recorded pitches (radar system). Accuracy was confirmed by ensuring that the times of throws identified in the sensor data and the times of throws identified by the radar system, which were matched for the duration of the bullpen session. 

We calculated workload from acceleration data using four different methods. One was a simple acceleration metric that could easily be calculated from raw inertial sensor data (i.e., peak resultant acceleration), two were calculations popularized by consumer wearable systems (i.e., peak PlayerLoad and cumulative PlayerLoad), and one was a relative metric of acceleration (i.e., normalized resultant acceleration). In this paper, we use the term “workload” to denote calculated estimates of external load, which is a potential proxy for the mechanical load experienced. Peak resultant acceleration (WL1) was calculated from the tri-axial acceleration signals. PlayerLoad per pitch was calculated using the equation given in Bullock et al. [11], which defines PlayerLoad as the resultant vector of the instantaneous rate of change in the direction of the tri-axial acceleration divided by a scaling factor: $$PlayerLoad=\sqrt{\frac{\Delta a^{2}+\Delta b^{2}+\Delta c^{2}}{100}\Delta a^{2}+\Delta b^{2}+\Delta c^{2}}$$ In their paper [11], Δa is the instantaneous change in anterior-posterior accelerations, Δb is the change in medial-lateral accelerations, and Δc is the change in the vertical accelerations. We used the x-y-z accelerations from each sensor to calculate peak PlayerLoad (WL2). Cumulative PlayerLoad for each pitch (WL3) was calculated in accordance with previous work [11]. In Bullock et al. [11], the cumulative PlayerLoad was calculated by summing the PlayerLoad across the duration of the pitching movement. The authors [11] provide no details about how the start and end of pitching motion was defined by the measured data. Because of this ambiguity, we defined the pitch as starting 0.5 s before peak upper arm acceleration and ending 0.5 s after peak upper arm acceleration. Finally, we calculated normalized resultant acceleration (WL4) by dividing the absolute tri-axial resultant acceleration by the maximum value experienced during the entire bullpen session for each pitcher. That is, the largest measured resultant acceleration from each sensor for the pitcher during the session equaled 1.0, and all others were denoted as a proportion of 1.0. Normalization was performed to better compare across players and pitch types. In addition, normalization helps with interpretation because absolute values of accelerations are expected to be sensitive to how the sensor is affixed, which can easily vary between pitchers and from day-to-day, even for the same pitcher. 

### 2.3. Statistical Analysis

A series of generalized linear mixed model analyses were used to determine the sensor and pitch type effect on workload estimates. We used Tukey’s method to make pairwise comparisons when significant main effects were found. A series of Pearson Correlation tests were performed to determine the association between the different workload estimates at each sensor location and for each pitch type. The alpha level was set a priori to 0.05. 

## 3. Results

Sensor location significantly influenced all workload estimates (*p* < 0.001) (Figure 2). Pairwise comparisons revealed that, regardless of calculation method, all estimates were significantly different between forearm and upper arm (*p* < 0.001), forearm and trunk (*p* < 0.001), and upper arm and trunk (*p* < 0.001). An interaction between sensor location and pitch type was found. Within sensor location, all types of workload estimates using forearm location data were significantly different across pitch type. Only cumulative PlayerLoad was significantly different across pitch types when using the upper arm location data (Table 2). No differences across pitch type were found using trunk location data. The significance of association between different workload estimates varied across pitch type (Table 3). Pitch performance metrics reported from the consumer-available radar unit (Pitching 2.0, Rapsodo, Inc.) can be found in Table 4.

## 4. Discussion

Two key findings of this study are that (1) sensor location matters when estimating external workload in baseball pitching and (2) workload estimates are not interchangeable. The forearm location appears to be much more sensitive to differences across pitch type, regardless of calculation method. Additionally, workload estimates using different calculations from acceleration data varied in their association to each other. This indicates that estimates may represent different concepts related to the external workload that a player experiences. Study findings highlight the need to be purposeful about the type of external load measurement desired and the method of collection. 

To date, only two studies have estimated workload during pitching using wearable sensors [6,11]. Mehta [6] collected workload using a commercially available wearable device (motusTHROW) for pre- and in-season throws from 18 baseball pitchers. Workload was quantified by using a forearm-mounted inertial sensor. The sensor outputted an estimate of elbow valgus torque, which is thought to be a key contributing factor to elbow ligamentous injury [10,12]. From this estimate, the acute-to-chronic valgus ratio (ACVR) was calculated using a method adapted from the acute-to-chronic workload ratio popularized by Gabbett et al. [13,14]. The second study also used a commercially available device to quantify workload, but from a trunk-mounted sensor [11]. While the raw workload data is not available from [6] for comparison, Bullock et al. [11] found that peak PlayerLoad from fastballs were significantly larger than change-ups (4.0 ± 0.9 versus 3.8 ± 0.9) and curveballs (4.2 ± 0.9 versus 4.1 ± 1.0), but not sliders (4.1 ± 1.1 versus 4.0 ± 1.0). Cumulative PlayerLoad differed significantly between fastballs and change-ups (270.3 ± 68.4 versus 257.9 ± 63.2), but not across other pitch types. These findings contrast with our own. We did not find significant differences between these pitch types using the trunk (or forearm) sensor data for either peak PlayerLoad or cumulative PlayerLoad. Differences in findings can be explained by differences in the subject population studied, the difference in sensor placement on the trunk (i.e., base of neck versus sternum), or differences in data processing, since filtering and algorithms from commercial systems are not disclosed. For example, while the specific calculation has not been made public, PlayerLoad appears to be derived from a scaled resultant of the instantaneous rate of change in each direction of the tri-axial acceleration, which is an estimation of the magnitude of ‘jerk,’ except that, instead of dividing by the change in time, a scaling factor is used. However, this method may not be robust or easily transferable, as the change in the acceleration per time step is sensitive to the accelerometer sampling rate. Bullock et al. [11] utilized a sampling rate of 100 Hz (where our study used 512 Hz) but did not state the relation between the scaling factor (100) and the sampling rate. [11] does not provide additional details about methodological choices based on their definition of PlayerLoad. Confusion about the definition and calculation methods of PlayerLoad is not unique, as other researchers have noted discrepancies [15]. 

Bullock et al [11] suggested that using workload estimates from a trunk-mounted sensor ‘could serve as a proxy [for shoulder or elbow loads] to begin to understand modifiable risk factors beyond pitch count that may help in reducing pitching injury. However, our findings indicate that trunk-based workload estimates may not appropriately reflect external mechanical load at the elbow (throwing forearm) or the shoulder (throwing upper arm). This is supported by previous work with optical motion capture where higher trunk velocity was linked to lower elbow joint torque [16,17,18]. Conceptually, the nature of the kinetic chain in the throwing motion moves energy along sequential segments (proximal-to-distal) such that high outputs from the trunk translate to lower mechanical stress at the arm. To date, a link between trunk acceleration or its derivative, jerk, and elbow stress is not well documented. Workload estimates from trunk-mounted sensors have largely been used in team sports with high volumes of multidirectional movements of varying amplitudes [19,20]. The Catapult OptimEye S5 system uses a trunk-mounted sensor to calculate PlayerLoad™. The rationale for this is that jerky movements (i.e., larger changes in instantaneous acceleration) of the center of mass result in more muscle activity and higher energy expenditure [21]. Peaks values can be thought of as the magnitude of the jerk and cumulative values as an estimate equivalent to the integral of the jerk magnitude. Given this calculation and sensor placement, PlayerLoad™ appears to be best-suited as a measure of whole-body external load, with its original application in running-based team sports like rugby, Australian Rules football, and global football. 

The desired load in baseball pitching is external load on the throwing arm, most importantly the forces acting on the elbow joint. The motusTHROW device may be more applicable since the sensor is located on the proximal forearm. It is important to note that motusTHROW has not been tested for day-to-day reliability, and its accuracy in estimating elbow valgus torque against a gold standard has deemed it, to date, as acceptable for ‘casual use’ only [22,23]. In our study, we placed the forearm sensor on the distal end of the segment for both comfort and accuracy of segment kinematics. While the sensor location in [6] was more accurate than [11] for assessing external pitching workloads, the workload estimate metric chosen was potentially problematic. Mehta [6] used daily workloads, computed as the sum of the estimated elbow valgus torque incurred for each throw, to calculate an acute-to-chronic varus ratio (ACVR). Importantly, acute to chronic windows were originally designed to assess physiological fitness versus fatigue in rugby athletes who practice (run) daily. It is unclear whether these windows are appropriate for pitching. Many pitchers have rotating bullpens and do not throw every day, particularly during the season. Likewise, adaptation and recovery timescales are different between musculoskeletal tissue and cardiac endurance. Thus, ACVR from [6] is unlikely to represent the true elbow ligamentous response to external loading. 

Findings from our study indicate that external workload at the elbow is different between fastballs, curveballs, and sliders, but not change-ups. For the player that threw cutters, this pitch appears to play a different workload in comparison to only sliders. While all estimates used the same acceleration signal, calculations differed and produced varying levels of association across pitch types. This suggests that calculation method, which represents the selection of a specific workload estimate, matters for assessing and monitoring pitching mechanical loads. At this point, it is unclear which estimate is most useful for forecasting injury or monitoring training adaptation. Additional work is needed to determine the estimate that most accurately reflects biological tissue load. However, we expect that peak resultant acceleration from the pitch cycle will be among the primary workload estimates used because both reaction forces and moments at the elbow joint during baseball pitching are dominated by the acceleration of the forearm [24], and because the sensor accelerometer range (±200 g) is more than sufficient to fully capture the large dynamics of baseball pitching. Additionally, normalizing this acceleration may be useful to assist with pitcher-specific monitoring and training. In our study, we chose to calculate normalized peak resultant accelerations to the peak acceleration of the entire bullpen session because inertial sensors are sensitive to changes in alignment and location on the segment [25] as well as the secureness of the affixation to the segment [26]. Without normalization, inter-subject acceleration cannot be easily generalized and within-subject or between-session comparisons may also be limited. 

Normalized acceleration was significantly different between breaking balls and fastballs but not change-ups, which suggests that pitchers may experience the same relative workload when throwing fastballs and change-ups but higher workloads when throwing breaking balls (curveballs or sliders). This is somewhat contrasted to previous work with professional pitchers, where researchers suggested that injury risk was equally high for throwing fastballs, sliders, and curveballs but not change-ups because of similar shoulder and elbow forces and torques outputs during optical motion capture testing [27]. Similarly, previous work with collegiate pitchers reported that resultant joint loads were similar between fastballs and curveballs and deemed these pitch types to be risky. They interpreted low kinetics in the change-up to imply that this pitch type was safest [28]. Our findings indicate that change-ups, in particular, appear to influence cumulative upper arm workload estimates, whereas fastballs and curveballs had lower loads to the upper arm. Thus, change-ups may be more stressful to the shoulder than the elbow. We suggest that relative rather than absolute comparisons may be more helpful in determining which pitch types could negatively overload throwing arm structures. 

Several limitations exist for this study. Firstly, we did not have an equal distribution of pitch types across players. All pitchers threw fastballs, change-ups, and curveballs. However, only one pitcher in our sample threw sliders and only two threw cutters. Although statistically accounted for, these findings for sliders and cutters should be interpreted cautiously. Secondly, this is a cross-sectional study, so no inference to injury risk can be made. Furthermore, the link between forearm acceleration, elbow joint load, and elbow injury remains to be definitively established. We did not directly calculate elbow valgus torque. While elbow valgus torque is the most often cited parameter to estimate joint stress, it requires an accurate measure of rotational forearm velocity, rotational forearm acceleration, linear forearm acceleration, forearm orientation, and upper arm orientation, as well as accurate estimates of body segment parameters and sensor placement relative to the elbow joint center [24]. Our inertial sensor had the capacity to capture rotations up to 2000°/s. However, forearm angular rates during pitching have been reported to be between 6000°/s to 9000°/s. We did not want to risk erroneous outputs due to the saturation of the angular velocity signal, and instead opted to use resultant acceleration [24]. Finally, we did not take any anthropometric measures of body segments or player heights. Acceleration may be influenced by body segment dimensions and, thus, should be explored in future work. 

## 5. Conclusions

The forearm location was most sensitive to differences in workload across pitch types, regardless of calculation method. Importantly, there is a need for standardization of operational definitions related to workload concepts and methods of calculation. In the meantime, multiple workload estimates should be calculated, including normalized estimates, since they can help determine player- and pitch-specific workloads and assist with individualized training or rehabilitation programs. Future research investigating the sensor-derived workload estimates should incorporate longitudinal observation and the collection of tissue-specific data like medical imaging.

## Figures and Tables

**Figure 1 sensors-22-09008-f001:**
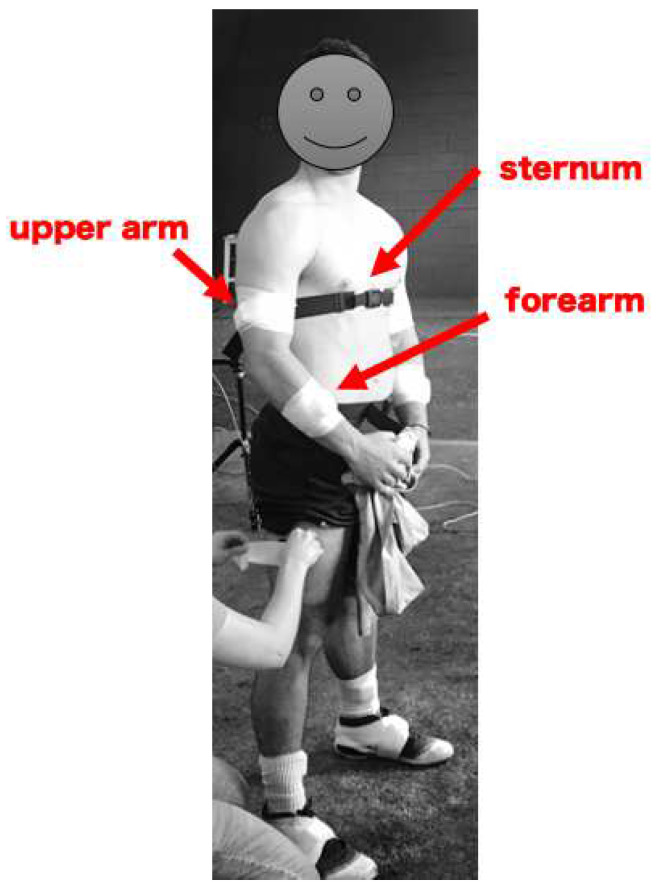
Illustration of sensor placement on the forearm, upper arm, trunk, and feet. The trunk sensor was placed on the sternum at the level of the xyphoid process, the upper arm sensor was placed at the posterolateral portion of the distal one-third of the segment, and the forearm sensor was placed at the mid-point of the segment. Placement location on each segment was determined by comfort for the pitcher and the location that appeared to produce minimal migration due to muscle contraction across multiple pitches. Arm and foot sensors were secured with flexible adhesive tape, while the trunk sensor was held in place by a snug chest strap.

**Figure 2 sensors-22-09008-f002:**
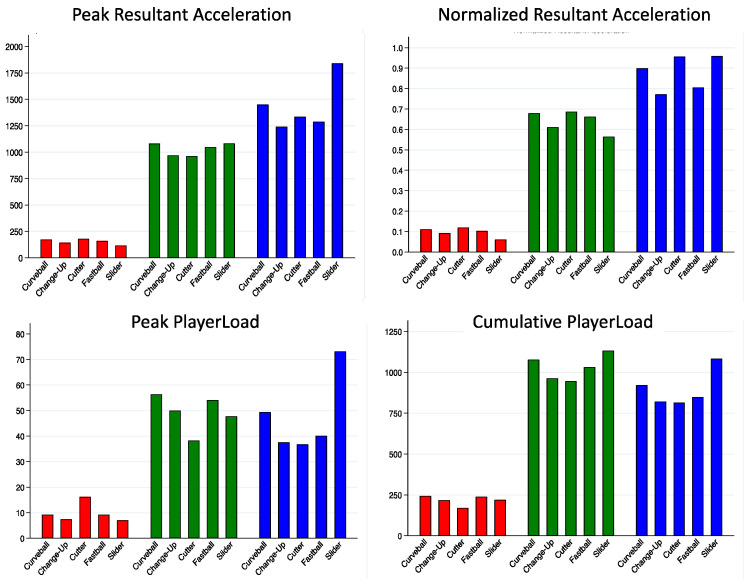
Mean workload estimates across pitch type and sensor location. Red indicates values from sternum sensor; Green indicates values from upper arm sensor; and Blue indicates values from forearm sensor. Workload estimates varied by calculation method and did not show the same pattern across pitch types.

**Table 1 sensors-22-09008-t001:** Total throw count for each pitch type by player. Warm-up throws are included to provide a more comprehensive picture of the training session but were not included in analysis.

Player	Warm-Up Throws	Fastballs	Change-Ups	Curveballs	Sliders	Cutters	Total
1	25	18	7	10	-	-	60
2	17	21	13	16	-	-	67
3	20	19	7	6	5	-	57
4	17	19	7	10	-	-	53
5	12	17	7	10	-	-	46
6	20	18	7	9	-	-	54
7	8	18	7	10	-	-	43
8	16	17	7	6	-	4	50
9	24	19	7	8	-	2	60
10	17	18	7	10	-	-	52

**Table 2 sensors-22-09008-t002:** Workload Estimates Across Pitch Types and Sensor Locations. Pitch type included fastballs (n =184), change-ups (n = 76), curveballs (n = 95), sliders (n = 5), and cutters (n = 6). Peak resultant acceleration is expressed in m/s^2^; all other estimates are unitless.

	Peak Resultant Acceleration (WL1)	Peak PlayerLoad (WL2)	Cumulative PlayerLoad (WL3)	Normalized Resultant Acceleration (WL4)
Trunk				
Fastball	160.5 ± 113.4	9.2 ± 12.3	238.6 ± 87.6	0.10 ± 0.08
Change-Up	143.3 ± 84.1	7.4 ± 8.7	217.3 ± 85.2	0.09 ± 0.06
Curveball	172.6 ± 118.0	9.2 ± 11.9	243.8 ± 87.9	0.11 ± 0.08
Slider	17.2 ± 102.8	7.0 ± 6.8	219.7 ± 201.3	0.06 ± 0.05
Cutter	180.2 ± 266.2	16.2 ± 25.3	170.7 ± 230.9	0.12 ± 0.18
	*p* = 0.34	*p* = 0.57	*p* = 0.73	*p* = 0.34
Upper Arm				
Fastball	1046.6 ± 184.0	54.1 ± 17.0	1031.4 ± 166.5	0.66 ± 0.14
Change-Up	969.8 ± 209.6	50.0 ± 20.8	963.5 ± 186.8 *	0.61 ± 0.15
Curveball	1082.5 ± 194.4	56.3 ± 23.5	1077.8 ± 157.3 **	0.68 ± 0.12
Slider	1082.9 ± 45.2	47.8 ± 3.2	1132.6 ± 69.0	0.56 ± 0.02
Cutter	961.9 ± 123.5	38.2 ± 10.1	945.7 ± 59.4	0.69 ± 0.04
	*p* = 0.36	*p* = 0.65	*p* = 0.04	*p* = 0.51
Forearm				
Fastball	1288.7 ± 184.1	40.1 ± 12.1	848.9 ± 97.0	0.81 ± 0.08
Change-Up	1241.3 ± 190.6	37.6 ± 12.3	820.8 ± 126.5	0.77 ± 0.07
Curveball	1450.9 ± 243.5 *^,^**	49.4 ± 17.1 *^,^**	921.2 ± 104.8 *^,^**	0.90 ± 0.05 *^,^**
Slider	1840.5 ± 82.0 *^,^**	73.2 ± 5.1 *^,^**^,^***	1083.3 ± 56.7 *^,^**^,^***	0.96 ± 0.04 *^,^**^,^***
Cutter	1336.3 ± 116.2 ****	36.7 ± 6.1 ****	814.9 ± 81.8 ****	0.96 ± 0.03 ****
	*p* < 0.001	*p* = 0.05	*p* = 0.008	*p* < 0.001

* significantly different from fastball, ** from change-up, *** from curveball, and **** from slider (*p* < 0.05).

**Table 3 sensors-22-09008-t003:** Correlation Among Workload Estimates Across Pitch Type. Only forearm sensor location data is used here. WL1 = peak resultant acceleration; WL2 = peak PlayerLoad WL3 = cumulative PlayerLoad; WL4 = normalized resultant acceleration.

Fastball					Change-Up				Curveball				Slider				Cutter			
	WL1	WL2	WL3	WL4	WL1	WL2	WL3	WL4	WL1	WL2	WL3	WL4	WL1	WL2	WL3	WL4	WL1	WL2	WL3	WL4
WL1	1.00				1.00				1.00				1.00				1.00			
WL2	**0.86**	1.00			**0.78**	1.00			**0.92**	1.00			−0.03	1.00			**0.89**	1.00		
WL3	**0.59**	**0.54**	1.00		**0.66**	**0.63**	1.00		**0.72**	**0.73**	1.00		0.36	−0.56	1.00		**0.96**	**0.94**	1.00	
WL4	0.24	0.06	**0.26**	1.00	**0.33**	0.10	**0.44**	1.00	**0.41**	**0.31**	**0.31**	1.00	**1.00**	−0.03	0.36	1.00	0.74	0.43	0.56	1.00

Note: bold and shaded indicates significant association (*p* < 0.05). Red shaded cells indicate a strong association (0.7 to 1.0) between measures, orange shaded cells indicate a moderate association (0.4 to 0.69), and yellow shaded cells indicate a weak (<0.4) association.

**Table 4 sensors-22-09008-t004:** Pitch Performance Metrics. Metrics displayed in mean ± SD and reporting directly from Rapsodo software program. Definitions are taken directly from the user manual (Rapsodo, Inc., Appendix A). Speed (mph) indicates how fast a pitch is traveling during flight. Spin rate is the rate at which the ball spins during flight. True spin is the spin directly impacting the movement of a pitch. Also known as “useful spin,” it is perpendicular to the direction the ball is traveling, deflecting the otherwise straight horizontal and vertical path of the ball. Spin efficiency is the percentage of spin directly impacting the movement of a pitch; spin efficiency is the ratio of true spin to total spin. Horizontal and vertical break represent how much a ball has moved when it crosses the strike zone compared to what its position would have been without spin.

	Speed (mph)	Spin (rpm)	True Spin (rpm)	Spin Efficiency	Horizontal Break	Vertical Break
Fastball	83.4 ± 4.4	1339.9 ± 956.9	1174.0 ± 849.8	58.7 ± 41.8	−0.22 ± 8.9	9.45 ± 8.4
Change-Up	76.6 ± 5.3	1171.6 ± 752.5	1009.9 ± 674.2	62.2 ± 39.8	0.39 ± 11.2	7.65 ± 6.1
Curveball	72.7 ± 4.3	1531.6 ± 990.0	857.0 ± 620.7	40.5 ± 29.9	−1.24 ± 9.2	−3.63 ± 6.0
Slider	81.0 ± 1.4	2150.0 ± 37.2	558.2 ± 57.8	26.0 ± 2.8	−2.82 ± 0.47	6.62 ± 1.1
Cutter	72.3 ± 4.1	959.7 ± 1055.1	339.5 ± 372.9	17.7 ± 19.4	−0.28 ± 4.3	1.75 ± 4.4
Grand Total	79.0 ± 6.5	1359.5 ± 932.6	1035.9 ± 765.7	53.6 ± 39.3	−0.39 ± 9.4	5.52 ± 9.1

## Data Availability

The data that support the findings of this study are available from the corresponding author upon reasonable request.

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
