# Peer review of "Sensor Location Matters When Estimating Player Workload for Baseball Pitching"

_sensors, 2022, doi:10.3390/s22229008_

Round 1
Reviewer 1 Report
The paper contains an interested application of sensor in rehabilitation field. For that, authors provided a little experimental campaign to estimate player workload.
The organization of paper is not clear. The introduction is more short and the SOA is not accurate. The description of experimental campaign must be a section of manuscript. The results section is not clear, because it is only composed by table and figures.
The importance and timeliness of the topic addressed in the paper within its area of research is good. In conclusion, the topic of paper and experimental campaign are interested but the manuscript needs a deep revision.
Author Response
Point 1: The paper contains an interested application of sensor in rehabilitation field. For that, authors provided a little experimental campaign to estimate player workload.
Response 1: We have edited and re-organized the methods section to include more information about data collection and specific informaiton about warm-up and counts for pitch type.
Point 2: The organization of paper is not clear. The introduction is more short and the SOA is not accurate. The description of experimental campaign must be a section of manuscript. The results section is not clear, because it is only composed by table and figures.
Response 2: We have updated the methods sections with sub-headers to make the experimental procedures clearer to the reader. Text for the results section can be found from Lines 176 to 187.
Point 3: The importance and timeliness of the topic addressed in the paper within its area of research is good. In conclusion, the topic of paper and experimental campaign are interested but the manuscript needs a deep revision.
Response 3: We have used the comments from both reviewers to revise the entire mansucript.
Reviewer 2 Report
Below is my feedback
Introduction
1. The introduction is very well written with the authors citing the appropriate literature. However, they make a case that these wearables are worn in different locations without providing the reader any information as to where these wearables were worn in the studies that they cited. My suggestion would be to provide additional information regarding the placement of the wearables to allow the reader to compare and contrast the locations. This should also lead you well into your objective statement.
Methodology
I appreciate the thoroughness of the researchers in describing the methodology however, there are still some questions that should be answered in order to improve the replicability of the study.
1. Please provide a chart of the pitches and the pitch count for each pitch.
2. Was the warm-up standardized? If, so what was it?
3. How many pitchers threw additional pitches? Were those pitches included in the analysis?
4. What AOS technologies cameras were utilized and where were was the location? Was there just 1? More than 1?
5. Were you collecting data for APDM and the camera on the same computer? If not, did you sync the clocks prior to data collection?
6. This might be an editorial issue, but can you please move Figure 1 to the methodology section instead of keeping it in the results section.
7. Although most of my work with APDM sensors has been in lower extremity movements, did you normalize the data by player height? Would height impact the reading? My thought has more to do with arm length
Results
1. Since this is an OA journal with no charges for color, I would highly recommend the authors use colors as the light gray, gray and black in Figure 2 are a little hard to read
2. I'd also recommend using colors in Table 2. I think a heat map with colors accordingly would help make it easier to focus on the strength of the correlations
3. Other than that the results are well presented
Discussion
While the discussion is well written, it brings up some concerns in terms of what was reported in the methodology. Please make sure you clarify how many pitchers threw each type of pitch as this will help better interpret the results. Based on your writing of the methodology, it seems that each pitcher threw a set number of each type of pitch. Please clarify your methodology
I do appreciate the strength of the limitation.
There are also minor grammatical errors in the discussion
Can you also propose some future direction of this work?
Author Response
Point 1: [Introduction] The introduction is very well written with the authors citing the appropriate literature. However, they make a case that these wearables are worn in different locations without providing the reader any information as to where these wearables were worn in the studies that they cited. My suggestion would be to provide additional information regarding the placement of the wearables to allow the reader to compare and contrast the locations. This should also lead you well into your objective statement.
Response 1: We have added the location of the sensors from previous work to the introduction.
Point 2: [Methodolgy] I appreciate the thoroughness of the researchers in describing the methodology however, there are still some questions that should be answered in order to improve the replicability of the study.
1) Please provide a chart of the pitches and the pitch count for each pitch
2) Was the warm-up standardized? If, so what was it?
3) How many pitchers threw additional pitches? Were those pitches included in the analysis?
4) What AOS technologies cameras were utilized and where were was the location? Was there just 1? More than 1?
5) Were you collecting data for APDM and the camera on the same computer? If not, did you sync the clocks prior to data collection?
6) This might be an editorial issue, but can you please move Figure 1 to the methodology section instead of keeping it in the results section
7) Although most of my work with APDM sensors has been in lower extremity movements, did you normalize the data by player height? Would height impact the reading? My thought has more to do with arm length
Response 2: Based on your recommendations, we have edited the text and information in the methodology.
1) we have added a table of counts for each pitch type per player (Table 1)
2) we have added information about the warm-up process and time to complete. The warm-up was not standardized but similar acrocss pitchers.
3) we have added to the methods that 5 pitchers threw an additional 1-4 pitches and these were included in analysis.
4) We have added the model of the high-speed camera to the text and that we used 2 cameras placed in front of and to the side of the pitcher.
5) We did not collect the APDM data and video using the same computer. The APDM sensors were used in data logging mode, whereas the high speed camera video was captured through a direct connection the computer. Synchronization was not critical for the analysis in this manuscript. We synchronized the IMU and video data in post-processing by aligning the rapid increase of foot acceleration of the lead foot with the observed ground contact of the lead foot in the video.
6) We have moved this figure to the methods section.
7) We did not normalize data by player height. We used the definition of PlayerLoad from previous studies, which does not incorporate any player size information. Player load is essentially a measure of jerk (the derivative of acceleration). For taller pitchers, we would expect lower wrist accelerations than shorter pitches when throwing at the same speed, which would likely result in higher PlayerLoad values for the shorter pitchers. However, normalization of PlayerLoad to pitcher/player size and height should be explored in future studies. We also used the resultant acceleration for study, which we normalized using the largest recorded value -- this normalization removes any differences that may be due to player size/height or pitching style.
Point 3: [Results] 1) Since this is an OA journal with no charges for color, I would highly recommend the authors use colors as the light gray, gray and black in Figure 2 are a little hard to read, 2) I'd also recommend using colors in Table 2. I think a heat map with colors accordingly would help make it easier to focus on the strength of the correlations, 3) Other than that the results are well presented
Response 3: Thank you for this suggestion. We have added color to Figure 2 and Table 2 (now Table 3).
Point 4: [Discussion] While the discussion is well written, it brings up some concerns in terms of what was reported in the methodology. Please make sure you clarify how many pitchers threw each type of pitch as this will help better interpret the results. Based on your writing of the methodology, it seems that each pitcher threw a set number of each type of pitch. Please clarify your methodology. I do appreciate the strength of the limitation. There are also minor grammatical errors in the discussion. Can you also propose some future direction of this work?
Response 4: In the methods section, we have added a table that gives specific counts for each pitch type for each player. We have also added more information about the warm-up process for the picthers. We have reviewed the discussion and edited grammatical errors. We have included some directions of future work in our conclusion section.
Round 2
Reviewer 1 Report
The new version of manuscript is more clear. Authors fellow the reviewers suggestions.
Reviewer 2 Report
I appreciate the authors making the suggested edits.